# Selective Convolutional Units: Improving CNNs via Channel Selectivity

## Abstract

Bottleneck structures with identity (e.g., residual) connection are now emerging popular paradigms for designing deep convolutional neural networks (CNN), for processing large-scale features efficiently. In this paper, we focus on the information-preserving nature of identity connection and utilize this to enable a convolutional layer to have a new functionality of *channel-selectivity*, i.e., re-distributing its computations to important channels. In particular, we propose Selective Convolutional Unit (SCU), a widely-applicable architectural unit that improves parameter efficiency of various modern CNNs with bottlenecks. During training, SCU gradually learns the channel-selectivity on-the-fly via the alternative usage of (a) *pruning* unimportant channels, and (b) *rewiring* the pruned parameters to important channels. The rewired parameters emphasize the target channel in a way that selectively enlarges the convolutional kernels corresponding to it. Our experimental results demonstrate that the SCU-based models without any post-processing generally achieve both model compression and accuracy improvement compared to the baselines, consistently for all tested architectures.

## 1 Introduction

Nowadays, convolutional neural networks (CNNs) have become one of the most effective approaches in various fields of artificial intelligence. With a growing interest of CNNs, there has been a lot of works on designing more advanced CNN architectures (Szegedy et al., 2015; Simonyan & Zisserman, 2014; Ioffe & Szegedy, 2015). In particular, the simple idea of adding *identity connection* in ResNet (He et al., 2016a) has enabled breakthroughs in this direction, as it allows to train substantially deeper/wider networks than before by alleviating existed optimization difficulties in previous CNNs. Recent CNNs can scale over a thousand of layers (He et al., 2016b) or channels (Huang et al., 2017b) without much overfitting, and most of these "giant" models consider identity connections in various ways (Xie et al., 2017; Huang et al., 2017b; Chen et al., 2017). However, as CNN models grow rapidly, deploying them in the real-world becomes increasingly difficult due to computing resource constraints. This has motivated the recent literature such as network pruning (Han et al., 2015; He et al., 2017; Liu et al., 2017; Neklyudov et al., 2017), weight quantization (Rastegari et al., 2016; Courbariaux & Bengio, 2016; Chen et al., 2018), adaptive networks (Teerapittayanon et al., 2016; Figurnov et al., 2017; Bolukbasi et al., 2017; Huang et al., 2018), and resource-efficient architectures (Huang et al., 2017a; Sandler et al., 2018; Ma et al., 2018).

For designing a resource-efficient CNN architecture, it is important to process succinct representations of large-scale channels. To this end, the identity connections are useful since they allow to reduce the representation dimension to a large extent while "preserving" information from the previous layer. Such *bottleneck* architectures are now widely used in modern CNNs such as ResNet (He et al., 2016a) and DenseNet (Huang et al., 2017b) for parameter efficiency, and many state-of-the-art mobile-targeted architectures such as SqueezeNet (Iandola et al., 2016), ShuffleNet (Zhang et al., 2017b; Ma et al., 2018), MoblileNet (Howard et al., 2017; Sandler et al., 2018), and CondenseNet (Huang et al., 2017a) commonly address the importance of designing efficient bottlenecks.

**Contribution.** In this paper, we propose Selective Convolutional Unit (SCU), a widely-applicable architectural unit for efficient utilization of parameters in particular as a bottleneck upon identity connection. At a high-level, SCU performs a convolutional operation to transform a given input. The main goal of SCU, however, is rather to *re-distribute* their computations only to selected channels

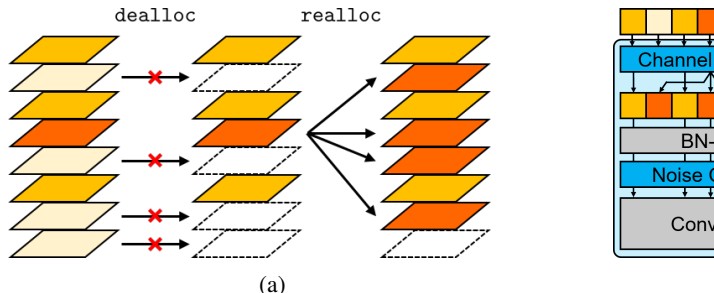

Figure 1: (a) An illustration of channel de-allocation and re-allocation procedures. The higher the saturation of the channel color, the higher the ECDS value. (b) The overall structure of SCU.

of importance, instead of processing the entire input naively. To this end, SCU has two special operations: (a) de-allocate unnecessary input channels (dealloc), and (b) re-allocate the obstructed channels to other channels of importance (realloc) (see Figure 1a). They are performed without damaging the network output (i.e., *function-preserving* operations), and therefore one can call them safely at any time during training. Consequently, training SCU is a process that increases the efficiency of CNN by iteratively pruning or rewiring its parameters on-the-fly along with learning them. In some sense, it is similar to how hippocampus in human brain learn, where new neurons are generated daily, and rewired into the existing network while maintaining them via neuronal apoptosis or pruning (Sahay et al., 2011a;b).

We combine several new ideas to tackle technical challenges for such on-demand, efficient trainable SCU. First, we propose *expected channel damage score* (ECDS), a novel metric of channel importance that is used as the criterion to select channels for dealloc or realloc. Compared to other popular magnitude-based metrics (Li et al., 2016; Liu et al., 2017; Neklyudov et al., 2017), ECDS allows capturing not only low-magnitude channels but also channels of low-contribution under the input distribution. Second, we impose *channel-wise spatial shifting bias* when a channel is reallocated, providing much diversity in the input distribution. It also has an effect of enlarging the convolutional kernel of SCU. Finally, we place a channel-wise scaling layer inside SCU with sparsity-inducing regularization, which also promotes dealloc (and consequently realloc as well), without further overhead in inference and training.

We evaluate the effectiveness of SCU by applying it to several modern CNN models including ResNet (He et al., 2016a), DenseNet (Huang et al., 2017b), and ResNeXt (Xie et al., 2017), on various classification datasets. Our experimental results consistently show that SCU improves the efficiency of bottlenecks both in model size and classification accuracy. For example, SCU reduces the error rates of DenseNet-40 model (without any post-processing) by using even less parameters: $6.57\% \rightarrow 5.95\%$ and $29.97\% \rightarrow 28.64\%$ on CIFAR-10/100 datasets, respectively. We also apply SCU to a mobile-targeted CondenseNet (Huang et al., 2017a) model, and further improve its efficiency: it even outperforms NASNet-C (Zoph et al., 2018), an architecture searched with 500 GPUs for 4 days, while our model is constructed with minimal efforts automatically via SCU.

There have been significant interests in the literature on discovering which parameters to be pruned during training of neural networks, e.g., see the literature of network sparsity learning (Wen et al., 2016; Lebedev & Lempitsky, 2016; Scardapane et al., 2017; Molchanov et al., 2017; Neklyudov et al., 2017; Louizos et al., 2017; 2018; Dai et al., 2018). On the other hand, the progress is, arguably, slower for how to rewire the pruned parameters of a given model to maximize its utility. Han et al. (2016) proposed Dense-Sparse-Dense (DSD), a multi-step training flow applicable for a wide range of DNNs showing that re-training with re-initializing the pruned parameters can improve the performance of the original network. Dynamic network surgery (Guo et al., 2016), on the other hand, proposed a methodology of splicing the pruned connections so that mis-pruned ones can be recovered, yielding a better compression performance. In this paper, we propose a new way of rewiring for parameter efficiency, i.e., rewiring for channel-selectivity, and a new architectural framework that enables both pruning and rewiring in a single pass of training without any post-processing or re-training (as like human brain learning). Under our framework, one can easily set a targeted trade-off between model compression and accuracy improvement depending on her purpose, simply by adjusting the calling policy of dealloc and realloc. We believe that our work sheds a new direction on the important problem of training neural networks efficiently.

## 2 SELECTIVE CONVOLUTIONAL UNITS

In this section, we describe Selective Convolutional Unit (SCU), a generic architectural unit for bottleneck CNN architectures. The overall structure of SCU is described in Section 2.1 and 2.2. In Section 2.3, we introduce a metric deciding channel-selectivity in SCU. We present in Section 2.4 how to handle a network including SCUs in training and inference.

### 2.1 OVERVIEW

**Bottleneck structures in modern CNNs.** We first consider a residual function defined in ResNet (He et al., 2016a) which has an identity mapping: for a given input random variable $\mathbf{X} \in \mathbb{R}^{I \times H \times W}$ ($H$ and $W$ are the height and width of each channel, respectively, and $I$ is the number of channels or feature maps) and a non-linear function $\mathcal{F}$, the output of a residual function is written by $\mathbf{Y} = \mathbf{X} + \mathcal{F}(\mathbf{X})$. This function has been commonly used as a building block for designing recent deep CNN models, in a form that $\mathcal{F}$ is modeled by a shallow CNN. However, depending on how $\mathcal{F}$ is designed, computing $\mathcal{F}(\mathbf{X})$ can be expensive when $I$ is large. For tackling the issue, *bottleneck structure* is a prominent approach, that is, to model $\mathcal{F}$ by $\mathcal{F}' \circ \mathcal{R}$ by placing a *bottleneck* $\mathcal{R}$ that firstly maps $\mathbf{X}$ into a lower dimension of $I' < I$ features. This approach, in essence, requires the identity connection, for avoiding information loss from $\mathbf{X}$ to $\mathbf{Y}$. Namely, the identity connection enables a layer to save redundant computation (or parameters) for just "keeping" information from the input. Bottleneck structures can be used other than ResNet as well, as long as the identity connection exists. Recent architectures including DenseNet (Huang et al., 2017b), PyramidNet (Han et al., 2017) and DPN (Chen et al., 2017) develop this idea with using a different aggregation function $\mathcal{W}$ instead of addition in ResNet, e.g., $\mathcal{W}(\mathbf{X}, \mathbf{X}') = [\mathbf{X}, \mathbf{X}']$ (channel-wise concatenation) for DenseNet. Designing $\mathcal{R}$-$\mathcal{F}'$-$\mathcal{W}$ is now a common way of handling large features.

**Channel-selectivity for efficient bottlenecks.** Although placing a bottleneck $\mathcal{R}$ reduces much computation of the main function $\mathcal{F}'$, we point out the majority of modern CNNs currently use inefficient design of $\mathcal{R}$ itself, so that even the computation of $\mathcal{R}$ often dominates the remaining. In ResNet and DenseNet models, for example, bottlenecks are designed using a pointwise convolution with a batch normalization layer (BN) (Ioffe & Szegedy, 2015) and ReLU (Nair & Hinton, 2010):

$$\mathcal{R} \leftarrow \mathrm{Conv}_{I \to I'}^{1 \times 1} \circ \mathrm{ReLU} \circ \mathrm{BN}, \tag{1}$$

where $\mathrm{Conv}_{I \to I'}^{1 \times 1}$ denotes a pointwise convolution that maps $I$ features into $I'$ features, i.e., its parameters can be represented by a $I \times I'$ matrix. This means that the parameters of $\mathcal{R}$ grows linearly on $I$, and it can be much larger than $\mathcal{F}'$ if $I \gg I'$. For example, in case of DenseNet-BC-190 (Huang et al., 2017b), 70% of the total parameters are devoted for modeling $\mathcal{R}$, which is inefficient as the expressivity of a pointwise convolution is somewhat limited. In this paper, we attempt to improve the efficiency of $\mathcal{R}$ in two ways: (a) reducing the parameters in $\mathrm{Conv}_{I \to I'}^{1 \times 1}$ by channel pruning, and (b) improving its expressivity by using the pruned parameters again. This motivates our goal to learn both channel-selectivity and parameters jointly.

**Overall architecture of SCU.** SCU is designed to learn the channel-selectivity via dynamic pruning and rewiring of channels during training. In this paper, we focus on putting SCU as a bottleneck $\mathcal{R}$, and show that the channel-selectivity of SCU improves its parameter efficiency. Our intuition is that (a) the information-preserving nature of identity connection brings optimization benefits if neurons in its structure are *dynamically* pruned during training, and (b) such pruning can be particularly effective on bottlenecks as their outputs are in a much lower dimension compared to the input. Nevertheless, we believe that our ideas on SCU are not limited to the bottleneck structures, as the concept of channel-selectivity can be generalized to other structures.

At a high level, SCU follows the bottleneck structure from (1), but for two additional layers: *Channel Distributor* (CD) and *Noise Controller* (NC) whose details are presented in Section 2.2. We model a non-linear function $\mathrm{SCU} : \mathbb{R}^{I \times H \times W} \to \mathbb{R}^{I' \times H \times W}$ as follows (see Figure 1b):

$$\mathcal{R} \leftarrow \mathrm{SCU} := \mathrm{Conv}_{I \to I'}^{1 \times 1} \circ \mathrm{NC} \circ \mathrm{ReLU} \circ \mathrm{BN} \circ \mathrm{CD}. \tag{2}$$

SCU has two special operations which control its input channels to process: (a) *channel de-allocation* (`dealloc`), which obstructs unnecessary channels from being used in future computations, and (b) *channel re-allocation* (`realloc`), which allocates more parameters to important,

non-obstructed channels by copying them into the obstructed areas. We design those operations to be *function preserving*, i.e. they do not change the original function of the unit, so that can be called at anytime during training without damage. Repeating `dealloc` and `realloc` alternatively during training translates the original input to what has only a few important channels, potentially duplicated multiple times. Namely, the parameters originally allocated to handle the entire input now operate on its important subset. On the way of designing the operations of function preserving, we propose *Expected Channel Damage Score* (ECDS) that leads to an efficient, safe way to capture unimportant channels by measuring how much the output of SCU changes on average (w.r.t. data distribution) after removing each channel. The details of ECDS are in Section 2.3.

## 2.2 DESIGN OF SCU: CD AND NC

**Channel Distributor (CD)** is the principal mechanism of SCU and is placed at the beginning of the unit. The role of CD is to "rebuild" the input, so that unnecessary channels can be discarded, and important channels are copied to be emphasized. In essence, we implement this function by re-indexing and blocking the input channel-wise: $\text{CD}(\mathbf{X})_i := g_i \cdot X_{\pi_i}$ with an index pointer $\pi_i \in \{1, 2, \cdots, I\}$, a gate variable $g_i \in \{0, 1\}$ for $i = 1, 2, \cdots, I$. Here, we notice that $\text{CD}(\mathbf{X})$ may contain a channel copied multiple times, i.e., multiple $\pi_i$'s can have the same value. Since SCU has different parameters for each channel, setting multiple $\pi_i$'s has an effect of allocating more parameters to better process the channel pointed by $\pi_i$.

We found that, however, it is hard to take advantage of the newly allocated parameters by simply copying a channel due to symmetry, i.e., the parameters for each channel usually degenerates. Due to this, we consider *spatial shifting biases* $\mathbf{b}_i = (b_i^h, b_i^w) \in \mathbb{R}^2$ for each channel, as illustrated in Figure 2. This trick can provide the copied channels much diversity in input distributions (and hence relaxing degeneracy), in a way that it is effective for the convolutional layer in SCU: it *enlarges* the convolutional kernel from $1 \times 1$ for the re-allocated channels only.

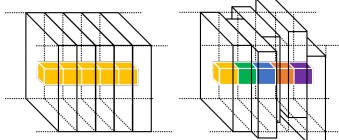

Figure 2: The kernel enlarging effect of spatial shifting.

To summarize, $\text{CD}(\mathbf{X})_i$ takes (a) the channel which $\pi_i$ is pointing, in the spatially shifted form with bias $\mathbf{b}_i$, or (b) **0** if the gate $g_i$ is closed. Formally, CD can be represented by $(\boldsymbol{\pi}, \mathbf{g}, \mathbf{b}) = (\pi_i, g_i, \mathbf{b}_i)_{i=1}^I$. $\text{CD}(\mathbf{X})$ has the same size to $\mathbf{X}$, and defined as follows:

$$\text{CD}(\mathbf{X})_i := g_i \cdot \text{shift}(X_{\pi_i}, b_i^h, b_i^w). \tag{3}$$

Here, $\text{shift}(X, b^h, b^w)$ denotes the "shifting" operation along spatial dimensions of $X$. For each pixel location $(i, j)$ in $X$, we define $\text{shift}(X, b^h, b^w)_{i,j}$ as:

$$\text{shift}\left(X, b^h, b^w\right)_{i,j} := \sum_{n=1}^{H} \sum_{m=1}^{W} X_{n,m} \cdot \max\left(0, 1 - |i - n + b^h|\right) \cdot \max\left(0, 1 - |j - m + b^w|\right) \tag{4}$$

using a bilinear interpolation kernel. This formulation allows $b^h$ and $b^w$ to be continuous real values, thereby to be learned via gradient-based methods with other parameters jointly.

**Noise Controller (NC)** is a component for more effective training of SCU. As SCU continuously performs channel pruning via `dealloc` during training, the efficiency of SCU depends on which regularization is used. The key role of NC is to induce the training of SCU to get more channel-wise sparsity, so that more channels can be de-allocated safely. Formally, NC is a channel-wise re-scaling layer: $\text{NC}(\mathbf{X}) := \mathbf{X} \odot \boldsymbol{\theta}$,[1] where $\boldsymbol{\theta} = (\theta_i)_{i=1}^I$ are parameters to be learned. For the channel-wise sparsity, we impose sparsity-inducing regularization specifically on $\boldsymbol{\theta}$.

Although any sparsity-inducing regularization can be used for $\boldsymbol{\theta}$ (Liu et al., 2017; Wen et al., 2016), in this paper we adopt the *Bayesian pruning* approach proposed by Neklyudov et al. (2017)[2] for two reasons: (a) it is easy to incorporate into training process, and (b) we found that noise incurred from Bayesian parameters helps to recover damage from channel pruning. In general, a Bayesian scheme regards each parameter $\theta$ as a random variable with prior $p(\boldsymbol{\theta})$. Updating the posterior $p(\boldsymbol{\theta}|\mathcal{D})$ from data $\mathcal{D}$ often leads the model to have much sparsity, if $p(\boldsymbol{\theta})$ is set to induce sparsity, e.g., by

---

[1] $\odot$ denotes the element-wise product.

[2] For completeness, we present for the readers an overview of Neklyudov et al. (2017) in Appendix B.

log-uniform prior (Kingma et al., 2015). Meanwhile, $p(\boldsymbol{\theta}|\mathcal{D})$ is usually approximated with a simpler model $q_{\boldsymbol{\phi}}(\boldsymbol{\theta})$, where $\phi$ are parameters to be learned. In case of NC, we regard each scaling parameter as a random variable, so that they become channel-wise multiplicative noises on input. We follow Neklyudov et al. (2017) for the design choices on $q_{\boldsymbol{\phi}}(\boldsymbol{\theta})$ and $p(\boldsymbol{\theta})$, by fully-factorized log-normal distribution $q_{\boldsymbol{\phi}}(\boldsymbol{\theta}) = \prod_i \mathrm{LogN}(\theta_i|\mu_i, \sigma_i^2)$ and log-uniform prior $p(\boldsymbol{\theta}) = \prod_i \mathrm{LogU}(\theta_i) \propto \frac{1}{\theta_i}$.

## 2.3 METRIC FOR CHANNEL-SELECTIVITY: ECDS

Consider an input random variable $\mathbf{X} = (X_i \in \mathbb{R}^{H \times W})_{i=1}^I$ of $I$ features. From now on, we denote a SCU by $\mathbf{S} = (W^{\mathrm{NC}}, W^{\mathrm{CD}}, W^{\mathrm{BN}}, W^{\mathrm{Conv}})$, where each denotes the parameters in NC, CD, BN, and Conv, respectively. Here, $W^{\mathrm{BN}} = (\gamma_i, \beta_i)_{i=1}^I$, and $W^{\mathrm{Conv}} = (W_i^{\mathrm{Conv}} \in \mathbb{R}^{I' \times 1 \times 1})_{i=1}^I$.

**Expected channel damage score (ECDS)** aims for measuring $\mathbb{E}[\mathrm{SCU}(\mathbf{X}; \mathbf{S}) - \mathrm{SCU}(\mathbf{X}; \mathbf{S}_{-i})] \in \mathbb{R}^{I' \times H \times W}$, where $\mathbf{S}_{-i}$ denotes a SCU identical to $\mathbf{S}$ but $g_i = 0$. In other words, it is the expected amount of changes in outputs when $\mathbf{S}_i$ is "damaged" or "pruned". The primary goal of this criteria is to make dealloc to be function preserving. We define $\mathrm{ECDS}(\mathbf{S})_i$ by the (Euclidean) norm of the averaged values of $\mathbb{E}[\mathrm{SCU}(\mathbf{X}; \mathbf{S}) - \mathrm{SCU}(\mathbf{X}; \mathbf{S}_{-i})]$ over the spatial dimensions:

$$\mathrm{ECDS}(\mathbf{S})_i := \left\| \frac{1}{HW} \sum_{h,w} \mathbb{E}[\mathrm{SCU}(\mathbf{X}; \mathbf{S}) - \mathrm{SCU}(\mathbf{X}; \mathbf{S}_{-i})]_{:,h,w} \right\|. \tag{5}$$

Notice that the above definition requires a marginalization over random variable $\mathbf{X}$. One can estimate it via Monte Carlo sampling using training data, but this is computationally too expensive compared to other popular magnitude-based metrics (Li et al., 2016; Liu et al., 2017; Neklyudov et al., 2017). Instead, we utilize the BN layer inside SCU, to infer the current input distribution of each channel at any time of training. This trick enables to approximate $\mathrm{ECDS}(\mathbf{S})_i$ by a closed formula of $\mathbf{S}_i$, avoiding expensive computations of $\mathrm{SCU}(\mathbf{X}; \cdot)$, as in what follows.

Consider a hidden neuron $x$ following BN and ReLU, i.e., $y = \mathrm{ReLU}(\mathrm{BN}(x))$, and suppose one wants to estimate $\mathbb{E}[y]$ *without* sampling. To this end, we exploit the fact that BN already "accumulates" its input statistics continuously during training. Under assuming that $\mathrm{BN}(x) \sim \mathcal{N}(\beta, \gamma^2)$ where $\gamma$ and $\beta$ are the scaling and shifting parameter in BN, respectively, it is elementary to check:

$$\mathbb{E}[y] = \mathbb{E}[\mathrm{ReLU}(\mathrm{BN}(x))] = |\gamma|\phi_{\mathcal{N}}\left(\frac{\beta}{|\gamma|}\right) + \beta\Phi_{\mathcal{N}}\left(\frac{\beta}{|\gamma|}\right), \tag{6}$$

where $\phi_{\mathcal{N}}$ and $\Phi_{\mathcal{N}}$ denote the p.d.f. and the c.d.f. of the standard normal distribution, respectively. The assumption is quite reasonable during training BN as each mini-batch is exactly normalized before applying the scaling and shifting inside BN. The idea is directly extended to obtain a closed form formula of $\mathrm{ECDS}(\mathbf{S})_i$ under some assumptions, as stated in the following proposition.

**Proposition 1.** *Assume* $\mathrm{BN}(\mathrm{CD}(\mathbf{X}; W^{\mathrm{CD}}); W^{\mathrm{BN}})_{i,h,w} \sim \mathcal{N}(\beta_i, \gamma_i^2)$ *for all* $i, h, w$.[3] *Then, it holds*

$$\mathrm{ECDS}(\mathbf{S})_i = g_i \cdot \underbrace{\left(|\gamma_i|\phi_{\mathcal{N}}\left(\frac{\beta_i}{|\gamma_i|}\right) + \beta_i\Phi_{\mathcal{N}}\left(\frac{\beta_i}{|\gamma_i|}\right)\right)}_{(a)} \cdot \underbrace{\mathbb{E}[\theta_i]}_{(b)} \cdot \underbrace{\|W_i^{\mathrm{Conv}}\|}_{(c)}, \qquad \text{for all } i.$$

The proof of the above proposition is given in Appendix D. In essence, there are three main terms in the formula: (a) a term that measures how much the input channel is active, (b) how much the NC amplifies the input, and (c) the total magnitude of weights in the convolutional layer. Therefore, it allows a way to capture not only low-magnitude channels but also channels of low-contribution under the input distribution (see Section 3.2 for comparisons with other metrics).

## 2.4 TRAINING AND INFERENCE PROCEDURES

Consider a CNN model $p(\mathbf{Y}|\mathbf{X}, \boldsymbol{\Theta})$ employing SCU, where $\boldsymbol{\Theta}$ denotes the collection of model parameters. For easier explanation, we rewrite $\boldsymbol{\Theta}$ by $(\mathbf{V}, \mathbf{W})$: $\mathbf{V}$ consists of $(\boldsymbol{\pi}, \mathbf{g})$ in CDs, and $\mathbf{W}$ is the remaining ones. Given dataset $\mathcal{D} = \{(x_n, y_n)\}_{n=1}^N$, $(\mathbf{V}, \mathbf{W})$ is trained via alternating two phases: (a) training $\mathbf{W}$ via stochastic gradient descent (SGD), and (b) updating $\mathbf{V}$ via dealloc

---

[3]In Appendix E, we also provide empirical supports on why the assumption holds in modern CNNs.

or `realloc`. The overall training process is mainly driven by (a), and the usage of (b) is optional. In (a), we use *stochastic variational inference* (Kingma & Welling, 2013) in order to incorporate stochasticity incurred from NC, so that SCU can learn its Bayesian parameters in NC jointly with the others via SGD. On the other hand, in (b), `dealloc` and `realloc` are called on demand during training depending on the purpose. For example, one may decide to call `dealloc` only throughout the training to obtain a highly compressed model, or one could use `realloc` as well to utilize more model parameters. Once (b) is called, (a) is temporally paused and $\mathbf{V}$ are updated.

**Training via stochastic variational inference.** We can safely ignore the effect of $\mathbf{V}$ during training of $\mathbf{W}$, since they remain fixed. Recall that, each noise $\theta$ from a NC is assumed to follow $q_\phi(\theta) = \mathrm{LogN}(\theta|\mu, \sigma^2)$. Then, $\theta$ can be "re-parametrized" with a noise $\varepsilon$ from the standard normal distribution as follows: $\theta = f(\phi = (\mu, \sigma), \varepsilon) = \exp(\mu + \sigma \cdot \varepsilon)$, where $\varepsilon \sim \mathcal{N}(0, 1^2)$. Stochastic variational inference (Kingma & Welling, 2013) allows a minibatch-based stochastic gradient method for $\theta$, in such case that $\theta$ can be re-parametrized with an non-parametric noise. The final loss we minimize for a minibatch $\{(x_{i_k}, y_{i_k})\}_{k=1}^M$ becomes (see Appendix F for more details):

$$\mathcal{L}_{\text{SCU}}(\mathbf{W}) = -\frac{1}{M} \sum_{k=1}^M \log p(y_{i_k}|x_{i_k}, f(\boldsymbol{\phi}, \boldsymbol{\varepsilon}_{i_k}), \mathbf{W}) + \frac{1}{N} \sum_{(\theta, \phi)} D_{\text{KL}}(q_\phi(\theta) \| p(\theta)) \qquad (7)$$

where $\boldsymbol{\varepsilon}_{i_k} = (\varepsilon_{i_k, u})_{u=1}^{|\boldsymbol{\phi}|}$ is a sampled vector from the fully-factorized standard normal distribution.

**Channel de-allocation and re-allocation.** Consider a SCU $\mathbf{S} = (W^{\text{NC}}, W^{\text{CD}}, W^{\text{BN}}, W^{\text{Conv}})$. The main role of `dealloc` and `realloc` is to update $W^{\text{CD}}$ in $\mathbf{S}$ that are not trained directly via SGD. They are performed as follows: *select* slices to operate by thresholding $\text{ECDS}(\mathbf{S})$, and *update* $\mathbf{S}$ from the selected channels. More formally, when `dealloc` is called, $\mathbf{S}_i$'s where $\text{ECDS}(\mathbf{S})_i < T_l$ for a fixed threshold $T_l$ are selected, and $g_i$'s in $W^{\text{CD}}$ are set by 0. If one chooses small $T_l$, this operation does not hurt the original function. On the other hand, `realloc` selects channels by collecting $\mathbf{S}_i$ where $\text{ECDS}(\mathbf{S})_i > T_h$, for another threshold $T_h$. Each of the selected channels can be re-allocated only if there is a closed channel in $\mathbf{S}$. If there does not exist a enough space, channels with higher ECDS have priority to be selected. A single re-allocation of a channel $\mathbf{S}_i$ to a closed channel $\mathbf{S}_j$ consists of several steps: (i) open $\mathbf{S}_j$ by $g_j \leftarrow 1$, (ii) copy $W_j^{\text{NC}}, W_j^{\text{BN}} \leftarrow W_i^{\text{NC}}, W_i^{\text{BN}}$ (iii) set $W_j^{\text{Conv}} \leftarrow \mathbf{0}$, (iv) re-initialize the shifting bias $\mathbf{b}_j$, and (v) set $\pi_j \leftarrow \pi_i$. This procedure is function-preserving, due to (iii).

After training a SCU $\mathbf{S}$, one can safely remove $\mathbf{S}_i$'s that are closed, to yield a compact unit. Then, CDs are now operated by "selecting" channels rather than by obstructing, thereby the subsequent layers play with smaller dimensions. Hence, at the end, SCU is trained to select only a subset of the input for performing the bottleneck operation. For NC, on the other hand, one can still use it for inference, but efficient inference can be performed by replacing each noise $\theta_i$ by constant $\mathbb{E}[\theta_i]$, following the well-known approximation used in many dropout-like techniques (Hinton et al., 2012).

## 3 EXPERIMENTAL RESULTS

In our experiments, we apply SCU to several well-known CNN architectures that uses bottlenecks, and perform experiments on CIFAR-10/100 (Krizhevsky, 2009) and ImageNet (Russakovsky et al., 2015) classification datasets. The more details on our experimental setups, e.g., datasets, training details, and configurations of SCU, are given in Appendix G.

### 3.1 IMPROVED CNN MODELS WITH SCU

**Improving existing CNNs with SCU.** We consider models using ResNet (He et al., 2016a), DenseNet (Huang et al., 2017b) and ResNeXt (Xie et al., 2017) architectures. In general, every model we used in this paper forms a stack of multiple bottlenecks, where the definition of each bottleneck differs depending on its architecture except that it can be commonly expressed by $\mathcal{R}$-$\mathcal{F}'$-$\mathcal{W}$ (the details are given in Table 5 in the appendix). We compare the existing models with the corresponding new ones in which the bottlenecks are replaced by SCU. For each SCU-based model, we consider three cases: (a) neither `dealloc` nor `realloc` is used during training, (b) only `dealloc` is used, and (c) both `dealloc` and `realloc` are used. We measure the total number of parameters in bottlenecks, and error rates.

Table 1: Comparison of performances on CIFAR-10/100 datasets in terms of their total parameter usage in $\mathcal{R}$-parts of the models, and their classification error rates. Here, "S" denotes whether SCU is used, and "D", "R" denote the use of `dealloc` and `realloc`, respectively. We indicate $k$ by the growth rate of DenseNet. All the values in the table are taken from averaging over 5 trials.

| Model | S | D | R | CIFAR-10 | | CIFAR-100 | |
|---|---|---|---|---|---|---|---|
| | | | | $\mathcal{R}$-Params | Error (%) | $\mathcal{R}$-Params | Error (%) |
| DenseNet-40 | ✗ | □ | □ | 0.11M | 6.57 | 0.11M | 29.97 |
| (bottleneck, $k=12$) | ✓ | ✗ | ✗ | 0.12M (+4.00%) | 6.30 (-4.11%) | 0.12M (+4.00%) | 29.25 (-2.40%) |
| | ✓ | ✓ | ✗ | **0.10M (-12.6%)** | 6.32 (-3.81%) | **0.11M (-2.84%)** | 29.31 (-2.20%) |
| | ✓ | ✓ | ✓ | 0.11M (-4.13%) | **5.95 (-9.44%)** | 0.11M (-0.95%) | **28.64 (-4.44%)** |
| DenseNet-100 | ✗ | □ | □ | 0.73M | 4.49 | 0.73M | 22.71 |
| (bottleneck, $k=12$) | ✓ | ✗ | ✗ | 0.76M (+4.11%) | 4.23 (-5.79%) | 0.76M (+4.11%) | 22.23 (-2.11%) |
| | ✓ | ✓ | ✗ | **0.36M (-51.4%)** | 4.41 (-1.78%) | **0.55M (-24.9%)** | 22.16 (-2.42%) |
| | ✓ | ✓ | ✓ | 0.60M (-18.2%) | **4.12 (-8.24%)** | 0.68M (-7.63%) | **21.34 (-6.03%)** |
| ResNet-164 (bottleneck) | ✗ | □ | □ | 0.39M | 4.23 | 0.39M | 21.28 |
| | ✓ | ✗ | ✗ | 0.41M (+5.13%) | 4.20 (-0.71%) | 0.41M (+5.13%) | 20.94 (-1.60%) |
| | ✓ | ✓ | ✗ | **0.20M (-48.0%)** | 4.10 (-3.07%) | **0.29M (-24.8%)** | 20.94 (-1.60%) |
| | ✓ | ✓ | ✓ | 0.32M (-19.3%) | **3.97 (-6.15%)** | 0.35M (-10.65%) | **20.49 (-3.71%)** |
| ResNeXt-29 ($4 \times 32d$) | ✗ | □ | □ | 1.72M | 4.05 | 1.72M | 19.82 |
| | ✓ | ✗ | ✗ | 1.73M (+0.05%) | 3.92 (-3.21%) | 1.73M (+0.05%) | 19.39 (-2.17%) |
| | ✓ | ✓ | ✗ | **1.09M (-33.8%)** | 3.96 (-2.22%) | **1.45M (-15.7%)** | 19.56 (-1.31%) |
| | ✓ | ✓ | ✓ | 1.42M (-17.2%) | **3.74 (-7.65%)** | 1.52M (-11.72%) | **19.17 (-3.28%)** |
| DenseNet-BC-190 ($k=40$) | ✗ | □ | □ | 17.5M | 2.72 | 17.5M | 16.20 |
| + *mixup* (Zhang et al., 2018) | ✓ | ✗ | ✗ | 17.7M (+1.22%) | 2.76 (+1.47%) | 17.7M (+1.22%) | **15.87 (-2.04%)** |
| | ✓ | ✓ | ✗ | **4.21M (-76.0%)** | 2.77 (+1.84%) | **7.12M (-59.4%)** | 16.26 (+0.37%) |
| | ✓ | ✓ | ✓ | 8.22M (-53.1%) | **2.69 (-1.10%)** | 12.4M (-29.0%) | 16.10 (-0.62%) |

Table 2: Comparison of model performances on ImageNet classification dataset. Here, we measure the single-crop validation error rates. "S" denotes whether the model uses SCU or not, and "D", "R" denote the use of `dealloc` and `realloc`, respectively.

| Model | S | D | R | $\mathcal{R}$-Params | Error (%) |
|---|---|---|---|---|---|
| ResNet-50 | ✗ | □ | □ | 3.50M | 23.19 |
| (bottleneck) | ✓ | ✗ | ✗ | 3.52M | 23.22 |
| | ✓ | ✓ | ✗ | **2.22M** | 23.19 |
| | ✓ | ✓ | ✓ | 3.04M | **22.82** |
| DenseNet-121 | ✗ | □ | □ | 1.00M | 23.63 |
| ($k=32$) | ✓ | ✗ | ✗ | 1.01M | 23.62 |
| | ✓ | ✓ | ✗ | **0.79M** | 23.62 |
| | ✓ | ✓ | ✓ | 0.97M | **23.24** |

Table 3: Comparison of performance on CIFAR-10 between different CNN models including ours: CondenseNet-SCU-182. Models named "X-Pruned" are the results by Liu et al. (2017).

| Model | Params | FLOPs | Error |
|---|---|---|---|
| ResNet-1001 | 16.1M | 2,357M | 4.62% |
| WRN-28-10 | 36.5M | 5,248M | 4.17% |
| ResNeXt-29 | 68.1M | 10,704M | 3.58% |
| DenseNet-190 | 25.6M | 9,388M | 3.46% |
| NASNet-C | 3.1M | - | 3.73% |
| VGGNet-Pruned | 2.30M | 391M | 6.20% |
| ResNet-164-Pruned | 1.10M | 275M | 5.27% |
| DenseNet-40-Pruned | 0.35M | 381M | 5.19% |
| CondenseNet-182 | 4.20M | 513M | 3.76% |
| **CondenseNet-SCU-182** | **2.59M** | **286M** | **3.63%** |

Table 1 compares the existing CNN models with the corresponding ones using SCU, on CIFAR-10/100. The results consistently demonstrate that SCU improves the original models, showing their effectiveness in different ways. When only `dealloc` is used, the model tends to be trained with minimizing their parameter to use. Using `realloc`, SCU now can utilize the de-allocated parameters to improve their accuracy aggressively. Note that SCU can improve the accuracy of the original model even neither `dealloc` nor `realloc` is used. This gain is from the regularization effect of stochastic NC, acting a dropout-like layer. We also emphasize that one can set a targeted trade-off between compression of SCU and accuracy improvement depending on her purpose, simply by adjusting the calling policy of `dealloc` and `realloc`. For example, in case of DenseNet-100 model on CIFAR-10, one can easily trade-off between reductions in (compression, error) = $(-51.4\%, -1.78\%)$ and $(-18.2\%, -8.24\%)$. In overall, SCU-based models achieve both model compression and accuracy improvement under all tested architectures. Table 2 shows the results on ImageNet, which are consistent to those on CIFAR-10/100. Notice that reducing parameters and error simultaneously is much more non-trivial in the case of ImageNet, e.g., reducing error $23.6\% \rightarrow 23.0\%$ requires to add 51 more layers to ResNet-101 (i.e., ResNet-152), as reported in the official repository of ResNet (He et al., 2016c).

**Designing efficient CNNs with SCU.** We also demonstrate that SCU can be used to design a totally new efficient architecture. Recall that, in this paper, SCU focus on the bottlenecks inside the overall

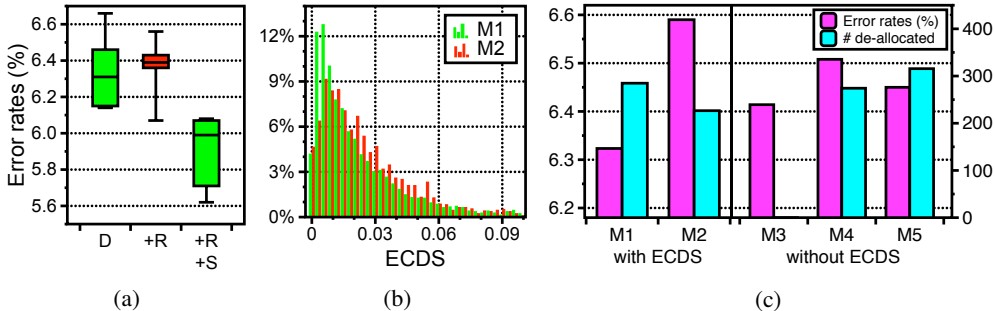

Figure 3: Ablation study on SCU. The model configurations for (b, c) are presented in Table 4. (a) Comparison of error rates between SCU (green) and the ablation on shifting (red). (b) Histograms on ECDS for all the channels of SCUs, using neither `dealloc` nor `realloc`. (c) Comparisons of the number of de-allocated channels and error rates. Here, all the models are trained with `dealloc`.

structure. The other parts, $\mathcal{F}'$ or $\mathcal{W}$, are other orthogonal design choices. To improve the efficiency of the parts, we adopt some components from CondenseNet (Huang et al., 2017a), which is one of the state-of-the-art architectures in terms of computational efficiency, designed for mobile devices. Although we do not adopt their main component, i.e., learned group convolution (LGC) as it also targets for the bottleneck as like SCU, we can still utilize other components of CondenseNet: increasing growth rate (IGR) (doubles the growth rate of DenseNet for every $N$ blocks starting from 8) and the use of group convolution for $\mathcal{F}'$. Namely, we construct a new model, coined *CondenseNet-SCU* by adopting IGR and GC upon a DenseNet-182 model with SCU. We replace each $3 \times 3$ convolution for $\mathcal{F}'$ by a group convolution of 4 groups. We train this model using `dealloc` only to maximize the computational efficiency. In Table 3, we compare our model with state-of-the-art level CNNs, including ResNet-1001 (He et al., 2016b), WRN-28-10 (Zagoruyko & Komodakis, 2016), NASNet-C (Zoph et al., 2018), and the original CondenseNet-182. As one can observe, our model shows better efficiency compared to the corresponding CondenseNet, suggesting the effectiveness of SCU over LGC. Somewhat interestingly, ours even outperforms NASNet-C that is an architecture searched over thousands of candidates, in both model compression and accuracy improvement. We finally remark that CondenseNet-SCU-182 model presented in Table 3 originally has 6.29M parameters in total before training, devoting 5.89M for bottlenecks, i.e., it is about 93.7% of the total number of parameters. This is indeed an example in that reducing overhead from bottlenecks is important for better efficiency, which is addressed by SCU.

## 3.2 ABLATION STUDY

We also perform numerous ablation studies on the proposed SCU, investigating the effect of the key components: CD, NC, and ECDS. For evaluation, we use the DenseNet-SCU-40 model (DenseNet-40 using SCU) trained for CIFAR-10. We also follow the training details described in Appendix G.

**Spatial shifting and re-allocation.** We propose spatial shifting as a trick in `realloc` procedure to provide diversity in input distributions. To evaluate its effect, we compare three DenseNet-SCU-40 models with different configurations of SCU: (D) only `dealloc` during training, (+R) `realloc` together but without spatial shifting, and (+R+S) further with the shifting. Figure 3a shows that +R does not improve the model performance much compared to D, despite +R+S outperforms both of them. This suggests that copying a channel naively is not enough to fully utilize the rewired parameters, and spatial shifting is an effective way to overcome the issue.

**Sparsity-inducing effect of NC.** We place NC in SCU to encourage more sparse channels. To verify such an effect, we consider DenseNet-SCU-40 model (say M1) and its variant removing NC from SCU (say M2). We first train M1 and M2 calling neither `dealloc` nor `realloc`, and compare them how the ECDS of each channel is distributed. Figure 3b shows that M1 tends to have ECDS closer to zero, i.e., more channels will be de-allocated than M2. Next, we train these models using `dealloc`, to confirm that NC indeed leads to more de-

Table 4: Configurations.

| Model | Metric | NC |
|---|---|---|
| M1 | ECDS | ☑ |
| M2 | ECDS | ☒ |
| M3 | SNR < 1 | ☑ |
| M4 | SNR < 2.3 | ☑ |
| M5 | $\ell_2 < 0.25$ | ☑ |

allocation. The left panel of Figure 3c shows that the number of de-allocated channels of M1 is relatively larger than that of M2, which is the desired effect of NC. Note that M1 also outperforms M2 on error rates, which is an additional advantage of NC from its stochastic regularization effect.

**Effectiveness of ECDS.** Nevertheless, remark that M2 in Figure 3c already de-allocates many channels, which suggests that SBP (used in NC) is not crucial for efficient de-allocation. Rather, the efficiency mainly comes from ECDS. To prove this claim, we evaluate three variants of M1 which use different de-allocation policies than $\text{ECDS} < T_l$: (a) $\text{SNR} < 1$ (thresholding the signal-to-noise ratio of NC in each channel by 1, proposed by the original SBP; M3), (b) $\text{SNR} < 2.3$ (M4) and (c) $\ell_2 < 0.25$ (thresholding $\|W_i^{\texttt{Conv}}\|_2$; M5). We train them using only $\texttt{dealloc}$, and compare the performances with the proposed model (M1). The right panel of Figure 3c shows the results of the three variants. First, we found that the M3 could not de-allocate any channel in our setting (this is because we prune a network on-the-fly during training, while the original SBP only did it after training). When we de-allocate competitive numbers of channels against M1 by tuning thresholds of others (M4 and M5), the error rates are much worse than that of M1. These observations confirm that ECDS is a more effective de-allocation policy than other magnitude-based metrics.

## 4 CONCLUSION

We demonstrate that CNNs of large-scale features can be trained effectively via channel-selectivity, primarily focusing on bottleneck architectures. The proposed ideas on channel-selectivity, however, would be applicable other than the bottlenecks, which we believe is an interesting future research direction. We also expect that channel-selectivity has a potential to be used for other tasks as well, e.g., interpretability (Selvaraju et al., 2017), robustness (Goodfellow et al., 2014), and memorization (Zhang et al., 2017a).

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

## A    Stochastic variational inference for Bayesian models

Consider a probabilistic model $p(\mathbf{Y}|\mathbf{X}, \boldsymbol{\theta})$ between two random variables $\mathbf{X}$ and $\mathbf{Y}$, and suppose one wants to infer $\boldsymbol{\theta}$ from a dataset $\mathcal{D} = \{(x_n, y_n)\}_{n=1}^N$ consisting $N$ i.i.d. samples from the distribution of $(\mathbf{X}, \mathbf{Y})$. In *Bayesian inference*, $\boldsymbol{\theta}$ is regarded as a random variable, under assuming some prior knowledge in terms of a prior distribution $p(\boldsymbol{\theta})$. The dataset $\mathcal{D}$ is then used to update the posterior belief on $\boldsymbol{\theta}$, namely $p(\boldsymbol{\theta}|\mathcal{D}) = p(\mathcal{D}|\boldsymbol{\theta})p(\boldsymbol{\theta})/p(\mathcal{D})$ from the Bayes rule. In many cases, however, computing $p(\boldsymbol{\theta}|\mathcal{D})$ through Bayes rule is intractable since it requires to compute intractable integrals. To address the issue, *variational inference* approximates $p(\boldsymbol{\theta}|\mathcal{D})$ by another parametric distribution $q_{\boldsymbol{\phi}}(\boldsymbol{\theta})$, and tries to minimize the KL-divergence $D_{\mathrm{KL}}(q_{\boldsymbol{\phi}}(\boldsymbol{\theta})\|p(\boldsymbol{\theta}|\mathcal{D}))$ between $q_{\boldsymbol{\phi}}(\boldsymbol{\theta})$ and $p(\boldsymbol{\theta}|\mathcal{D})$. Instead of directly minimizing it, one typically maximizes the *variational lower bound* $L(\boldsymbol{\phi})$, due to the following:

$$D_{\mathrm{KL}}(q_{\boldsymbol{\phi}}(\boldsymbol{\theta})\|p(\boldsymbol{\theta}|\mathcal{D})) = -L(\boldsymbol{\phi}) + \sum_{n=1}^N \log p(y_n|x_n), \tag{8}$$

$$\text{where} \quad L(\boldsymbol{\phi}) = \sum_{n=1}^N \mathbb{E}_{q_{\boldsymbol{\phi}}}[\log p(y_n|x_n, \boldsymbol{\theta})] - D_{\mathrm{KL}}(q_{\boldsymbol{\phi}}(\boldsymbol{\theta})\|p(\boldsymbol{\theta})). \tag{9}$$

In case of complex models, however, expectations in (9) are still intractable. Kingma & Welling (2013) proposed an unbiased minibatch-based Monte Carlo estimator for them, which can be used when $q_{\boldsymbol{\phi}}(\boldsymbol{\theta})$ is representable by $\boldsymbol{\theta} = f(\boldsymbol{\phi}, \boldsymbol{\varepsilon})$ with a non-parametric noise $\boldsymbol{\varepsilon} \sim p(\boldsymbol{\varepsilon})$. For a minibatch $\{(x_{i_k}, y_{i_k})\}_{k=1}^M$ of size $M$, one can obtain

$$L_D(\boldsymbol{\phi}) := \sum_{n=1}^N \mathbb{E}_{q_{\boldsymbol{\phi}}}[\log p(y_n|x_n, \boldsymbol{\theta})] \simeq \frac{N}{M} \sum_{k=1}^M \log p(y_{i_k}|x_{i_k}, f(\boldsymbol{\phi}, \boldsymbol{\varepsilon}_{i_k})) =: L_D^{\mathrm{SGVB}}(\boldsymbol{\phi}) \tag{10}$$

$$L(\boldsymbol{\phi}) \simeq L^{\mathrm{SGVB}}(\boldsymbol{\phi}) = L_D^{\mathrm{SGVB}}(\boldsymbol{\phi}) - D_{\mathrm{KL}}(q_{\boldsymbol{\phi}}(\boldsymbol{\theta})\|p(\boldsymbol{\theta})). \tag{11}$$

Now we can solve optimize $L(\boldsymbol{\phi})$ by stochastic gradient ascent methods, if $f$ is differentiable. For a model having non-Bayesian parameters, say $\mathbf{W}$, we can still apply the above approach by maximizing

$$L(\boldsymbol{\phi}, \mathbf{W}) = \sum_{n=1}^N \mathbb{E}_{q_{\boldsymbol{\phi}}}[\log p(y_n|x_n, \boldsymbol{\theta}, \mathbf{W})] - D_{\mathrm{KL}}(q_{\boldsymbol{\phi}}(\boldsymbol{\theta})\|p(\boldsymbol{\theta})), \tag{12}$$

where $\boldsymbol{\phi}$ and $\mathbf{W}$ can be jointly optimized under $L^{\mathrm{SGVB}}(\boldsymbol{\phi}, \mathbf{W}) \simeq L(\boldsymbol{\phi}, \mathbf{W})$.

## B    Structured Bayesian pruning

*Structured Bayesian pruning* (SBP) (Neklyudov et al., 2017) is a good example to show how stochastic variational inference can be incorporated into deep neural networks. The SBP framework assumes $\mathbf{X}$ to be an object of $I$ features, that is, $\mathbf{X} = (X_i)_{i=1}^I$. For example, $\mathbf{X} \in \mathbb{R}^{I \times H \times W}$ can be a convolutional input consisting $I$ channels, of the form $\mathbf{X} = (X_i \in \mathbb{R}^{W \times H})_{i=1}^I$ where $W$ and $H$ denote the width and the height of each channel, respectively. It considers a dropout-like layer with a noise vector $\boldsymbol{\theta} = (\theta_i)_{i=1}^I \sim p_{\mathrm{noise}}(\boldsymbol{\theta})$, which outputs $\mathbf{X} \odot \boldsymbol{\theta}$ of the same size as $\mathbf{X}$.[4] Here, $\boldsymbol{\theta}$ is treated as a random vector, and the posterior $p(\boldsymbol{\theta}|\mathcal{D})$ is approximated by a fully-factorized truncated log-normal distribution $q_{\boldsymbol{\phi}}(\boldsymbol{\theta})$:

$$q_{\boldsymbol{\phi}}(\boldsymbol{\theta}) = \prod_{i=1}^I q(\theta_i|\mu_i, \sigma_i) = \prod_{i=1}^I \mathrm{LogN}_{[\mathrm{a,b}]}(\theta_i|\mu_i, \sigma_i^2) \tag{13}$$

$$\mathrm{LogN}_{[\mathrm{a,b}]}(\theta_i|\mu_i, \sigma_i^2) \propto \mathrm{LogN}(\theta_i|\mu_i, \sigma_i^2) \cdot \mathbf{1}_{[\mathrm{a,b}]}(\log \theta_i), \tag{14}$$

where $\mathbf{1}_{[\mathrm{a,b}]}$ denotes the indicator function for the inveral $[\mathrm{a, b}]$.

---

[4] $\odot$ denotes the element-wise product.

Meanwhile, the prior $p(\boldsymbol{\theta})$ is often chosen by a fully-factorized log-uniform distribution, e.g., Sparse Variational Dropout (Molchanov et al., 2017), and SBP use the truncated version:

$$p(\boldsymbol{\theta}) = \prod_{i=1}^{I} p(\theta_i) = \prod_{i=1}^{I} \text{LogU}_{[a,b]}(\theta_i). \tag{15}$$

The reason why they use truncations for $q_{\boldsymbol{\phi}}(\boldsymbol{\theta})$ and $p(\boldsymbol{\theta})$ is to prevent $D_{\text{KL}}(q_{\boldsymbol{\phi}}(\boldsymbol{\theta})\|p(\boldsymbol{\theta}))$ to be improper. Previous works (Kingma et al., 2015; Molchanov et al., 2017) ignore this issue by implicitly regarding them as truncated distributions on a broad interval, but SBP treats this issue explicitly.

Note that, each $\theta_i \sim q_{\boldsymbol{\phi}}(\theta_i) = \text{LogN}(\theta_i|\mu_i, \sigma_i^2)$ in the noise vector $\boldsymbol{\theta}$ can be re-parametrized with a non-parametric uniform noise $\varepsilon_i \sim \mathcal{U}(\varepsilon|0, 1)$ by:

$$\theta_i = f(\mu_i, \sigma_i, \varepsilon_i) = \exp\left(\mu_i + \sigma_i \Phi^{-1}\left(\Phi(\alpha_i) + \varepsilon_i(\Phi(\beta_i) - \Phi(\alpha_i))\right)\right) \tag{16}$$

where $\alpha_i = \frac{a-\mu_i}{\sigma_i}$, $\beta_i = \frac{b-\mu_i}{\sigma_i}$, and $\Phi$ denotes the cumulative distribution function of the standard normal distribution. Now one can optimize $\boldsymbol{\phi} = (\boldsymbol{\mu}, \boldsymbol{\sigma})$ jointly with the weights $\mathbf{W}$ of a given neural network via stochastic variational inference described in Section A. Unlike Molchanov et al. (2017), SBP regards $\mathbf{W}$ as a non-Bayesian parameter, and the final loss $\mathcal{L}_{\text{SBP}}$ to optimize becomes

$$\mathcal{L}_{\text{SBP}}(\boldsymbol{\phi}, \mathbf{W}) = -\frac{N}{M} \sum_{k=1}^{M} \log p(y_{i_k}|x_{i_k}, f(\boldsymbol{\phi}, \boldsymbol{\varepsilon}_{i_k}), \mathbf{W}) + \alpha \cdot D_{\text{KL}}(q_{\boldsymbol{\phi}}(\boldsymbol{\theta})\|p(\boldsymbol{\theta})). \tag{17}$$

Here, the KL-divergence term is scaled by $\alpha$ to compensate the trade-off between sparsity and accuracy. In practice, SBP starts from a pre-trained model, and re-trains it using the above loss. Due to the sparsity-inducing behavior of log-uniform prior, $\boldsymbol{\theta}$ is forced to become more noisy troughout the re-training. Neurons with $\theta$ of signal-to-noise ratio (SNR) below 1 are selected, and removed after the re-training:

$$\text{SNR}(\theta_i) = \frac{\mathbb{E}\theta_i}{\sqrt{\mathbb{V}\theta_i}} = \frac{(\Phi(\sigma_i - \alpha_i) - \Phi(\sigma_i - \beta_i))/\sqrt{\Phi(\beta_i) - \Phi(\alpha_i)}}{\sqrt{\exp(\sigma_i^2)(\Phi(2\sigma_i - \alpha_i) - \Phi(2\sigma_i - \beta_i)) - (\Phi(\sigma_i - \alpha_i) - \Phi(\sigma_i - \beta_i))^2}}. \tag{18}$$

## C  BAYESIAN PRUNING AND IDENTITY CONNECTIONS

SCU requires "training-time removal" of input channels for the channel de-allocation and re-allocation to work. But usually, this process should be done carefully since it can make the optimization much difficult and put the network into a bad local minima. In particular, it occurs if we select channels to remove too aggressively. It is known that this issue becomes more pronounced in Bayesian neural networks (Sønderby et al., 2016; Molchanov et al., 2017; Neklyudov et al., 2017; Louizos et al., 2017), such as SBP we use in this paper. Recall the variational lower bound objective in (12), for Bayesian parameters $\boldsymbol{\phi}$ and non-Bayesian $\mathbf{W}$. If the gradient of the first term $\sum_{n=1}^{N} \mathbb{E}_{q_{\boldsymbol{\phi}}}[\log p(y_n|x_n, \boldsymbol{\theta}, \mathbf{W})]$ on the right-hand side does not contribute much on $\nabla_{\boldsymbol{\phi}}\mathcal{L}(\boldsymbol{\phi}, \mathbf{W})$, then $\boldsymbol{\phi}$ will be optimized mostly by $-\nabla_{\boldsymbol{\phi}}D_{\text{KL}}(q_{\boldsymbol{\phi}}(\boldsymbol{\theta})\|p(\boldsymbol{\theta}))$, that is, to follow the prior $p(\boldsymbol{\theta})$. Unfortunately, in practice, we usually observe this phenomena at the early stage of training, when $\mathbf{W}$ are randomly initialized. In that case then, $q_{\boldsymbol{\phi}}(\boldsymbol{\theta})$ will become $p(\boldsymbol{\theta})$ too fast because of the "uncertain" $\mathbf{W}$, thereby many channels will be pruned forever, in SBP for example.

This problem is usually dealt with in one of two ways: (a) using a pre-trained network as a starting point of $\mathbf{W}$ (Molchanov et al., 2017; Neklyudov et al., 2017), and (b) a "warm-up" strategy, where the KL-divergence term is rescaled by $\beta$ that increases linearly from 0 to 1 during training (Sønderby et al., 2016; Louizos et al., 2017). In this paper, however, neither methods are used, but instead we have found that the problem can be much eased with identity connections, as it can eliminate a possible cause of the optimization difficulty from removing channels: optimization difficulty from losing information as an input passes through a deep network. The presence of identity connection implies that the information of an input will be fully preserved even in the case when all the parameters in a layer are pruned. This may not be true in models without identity, for example, in VGGNet (Simonyan & Zisserman, 2014), one can see that the information of an input will be completely lost if any of the layers removes its entire channels. This suggests us that identity connections can be advantageous not only for scaling up the network architectures, but also for reducing the size of them.

## D    PROOF OF PROPOSITION 1

One can assume that $\mathbf{S}_i$ is open, i.e. $g_i = 1$, otherwise $\text{ECDS}(\mathbf{S})_i = 0$ by definition since $\mathbf{S} = \mathbf{S}_{-i}$. Say $\mathbf{Y} := \text{BN}(\text{CD}(\mathbf{X}; W^{\text{CD}}); W^{\text{BN}})$, and let $\text{Conv}_{I \to I'}^{1 \times 1, (i)} : \mathbb{R}^{1 \times H \times W} \to \mathbb{R}^{I' \times H \times W}$ be the $i$-th slice of $\text{Conv}_{I \to I'}^{1 \times 1}$, which can be defined by the convolution with $W_i^{\text{Conv}}$ so that $\text{Conv}_{I \to I'}^{1 \times 1}(\mathbf{X}; W^{\text{Conv}}) = \sum_{i=1}^{I} \text{Conv}_{I \to I'}^{1 \times 1, (i)}(X_{i,:,:}; W_i^{\text{Conv}})$. Then, we have:

$$\begin{aligned}
\text{SCU}(\mathbf{X}; \mathbf{S}) - \text{SCU}(\mathbf{X}; \mathbf{S}_{-i}) =: \mathbf{S}_i(\mathbf{X}) &= \text{Conv}_{I \to I'}^{1 \times 1, (i)}((\text{NC} \circ \text{ReLU} \circ \text{BN} \circ \text{CD})(\mathbf{X})_i) \\
&= \text{Conv}_{I \to I'}^{1 \times 1, (i)}(\text{NC}(\text{ReLU}(\mathbf{Y}))_i) \\
&= \text{Conv}_{I \to I'}^{1 \times 1, (i)}(\max(\mathbf{Y}, 0)_i \odot \boldsymbol{\theta}) \\
&= \text{Conv}_{I \to I'}^{1 \times 1, (i)}(\max(Y_{i,:,:}, 0) \cdot \theta_i) =: \text{Conv}_{I \to I'}^{1 \times 1, (i)}(Z_i).
\end{aligned} \tag{19}$$

Now, check that $\text{ECDS}(\mathbf{S}_i)$ becomes:

$$\text{ECDS}(\mathbf{S}_i) = \left\| \frac{1}{HW} \sum_{h,w} \mathbb{E}[\mathbf{S}_i(\mathbf{X})]_{:,h,w} \right\| = \left\| \frac{1}{HW} \sum_{h,w} \mathbb{E}[\text{Conv}_{I \to I'}^{1 \times 1, (i)}(Z_i)]_{:,h,w} \right\| \tag{20}$$

By the assumption that $Y_{i,h,w} \sim \mathcal{N}(\beta_i, \gamma_i^2)$ for all $h, w$, we get:

$$Z_{i,h,w} = \max(Y_{i,h,w}, 0) \cdot \theta_i, \quad \text{and} \tag{21}$$

$$\begin{aligned}
\mathbb{E}[Z_{i,h,w}] &= \mathbb{E}[\max(Y_{i,h,w}, 0) \cdot \theta_i] = \mathbb{E}[\max(Y_{i,h,w}, 0)] \cdot \mathbb{E}[\theta_i] \\
&= \left( |\gamma_i| \phi_{\mathcal{N}}\left( \frac{\beta_i}{|\gamma_i|} \right) + \beta_i \Phi_{\mathcal{N}}\left( \frac{\beta_i}{|\gamma_i|} \right) \right) \cdot \mathbb{E}[\theta_i] =: f(\mathbf{S}_i) \qquad \forall h, w
\end{aligned} \tag{22}$$

where $\phi_{\mathcal{N}}$ and $\Phi_{\mathcal{N}}$ denote the probability distribution function and the cumulative distribution function of the standard normal distribution.

Therefore, the desired formula for $\text{ECDS}(\mathbf{S}_i)$ can be obtained by using the linearity of expectation:

$$\begin{aligned}
\text{ECDS}(\mathbf{S}_i) &= \left\| \frac{1}{HW} \sum_{h,w} \mathbb{E}[\text{Conv}_{I \to I'}^{1 \times 1, (i)}(Z_i)]_{:,h,w} \right\| \\
&= \left\| \frac{1}{HW} \sum_{h,w} \left( \mathbb{E}\left[ \sum_{x=-\lfloor 1/2 \rfloor}^{\lfloor 1/2 \rfloor} \sum_{y=-\lfloor 1/2 \rfloor}^{\lfloor 1/2 \rfloor} W_{i,j,x,y}^{\text{Conv}} \cdot Z_{i,h+x,w+y} \right] \right)_{j=1}^{I'} \right\| \\
&= \left\| \frac{1}{HW} \sum_{h,w} (W_{i,j}^{\text{Conv}} \cdot \mathbb{E}[Z_{i,h,w}])_{j=1}^{I'} \right\| \\
&= f(\mathbf{S}_i) \cdot \left\| \frac{1}{HW} \sum_{h,w} (W_{i,j}^{\text{Conv}})_{j=1}^{I'} \right\| \\
&= f(\mathbf{S}_i) \cdot \left\| W_i^{\text{Conv}} \right\| = g_i \cdot f(\mathbf{S}_i) \cdot \left\| W_i^{\text{Conv}} \right\|.
\end{aligned} \tag{23}$$

This completes the proof of Proposition 1.

## E    EMPIRICAL SUPPORTS ON THE ASSUMPTION OF PROPOSITION 1

To validate whether the assumption $\text{BN}(\text{CD}(\mathbf{X}; W^{\text{CD}}); W^{\text{BN}})_{i,h,w} \sim \mathcal{N}(\beta_i, \gamma_i^2)$ holds in modern CNNs, we first observe that, once we ignore the effects from spatial shifting,[5] a necessary condition of the assumption is that $(X_{i,:,:})$ are *identically distributed* normal for a given channel $X_i$. This is because BN and CD do not change the "shape" of pixel-wise distributions of $X_i$. From this observation, we conduct a set of experiments focusing on a randomly chosen hidden layer in a DenseNet-40 model. We analyze the empirical distribution of the hidden activation incoming to the layer calculated from CIFAR-10 test dataset. Since the data consists of 10,000 samples, we get an hidden activation $\mathbf{X}^{\text{test}} \in \mathbb{R}^{10000 \times C \times 32 \times 32}$,[6] where $C$ denotes the number of channels of the input.

---

[5] However, we emphasize that one can safely ignore this effect, since it can be successfully bypassed in practice by padding the input with boundary pixels.

[6] Here, we get a tensor of $32 \times 32$ channels, since we choose a layer from the first block of the model.

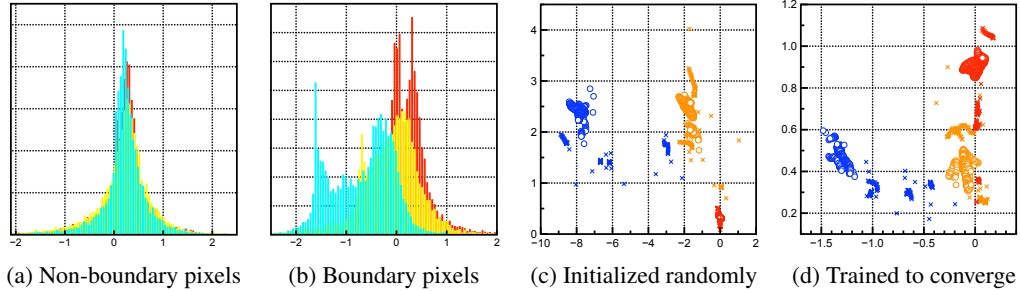

(a) Non-boundary pixels   (b) Boundary pixels   (c) Initialized randomly   (d) Trained to converge

Figure 4: Pixel-wise input distributions of a layer in DenseNet-40 model. (a, b) Empirical distributions of three randomly chosen pixels in a fixed channel of input, which are inferred from CIFAR-10 test dataset. (c, d) Scatter plots between empirical mean and standard deviation of each pixel distributions, plotted for 3 representative channels in the input. Each plot consists 1,024 points, as a channel have $32 \times 32$ pixels. Pixels in boundaries are specially marked as $\times$.

Now, suppose that we focus on a specific channel in $\mathbf{X}^{\texttt{test}}$, say $X_c^{\texttt{test}} \in \mathbb{R}^{10000 \times 32 \times 32}$. Notice that if our assumption is perfectly satisfied, then all the slices $X_{c,h,w}^{\texttt{test}} \in \mathbb{R}^{10000}$ will represent a fixed normal distribution for any $h, w$. Interestingly, by analyzing empirical distributions of $X_{c,h,w}^{\texttt{test}}$ for varying $h$ and $w$, we found that: (a) for a fixed $c$, most of the empirical distributions from $X_{c,h,w}^{\texttt{test}}$ have unimodal shapes, except for the pixels in boundaries of the channel (Figure 4a, 4b), and (b) for a large portion of $c \in \{1, \cdots, C\}$, the empirical means and variances of $X_{c,h,w}^{\texttt{test}}$'s are concentrated in a cluster (Figure 4c, 4d).

These observations suggest us that the assumption of Proposition 1 can be reasonable except for the boundaries. We also emphasize that these trends we found are also appeared even when the model is not trained at all (Figure 4c), that is, all the weights are randomly initialized, which implies that these properties are not "learned", but come from a structural property of CNN, e.g. equivariance on translation, or the central limit theorem. This observation provides us another support why the ECDS formula stated in Proposition 1 is valid at any time during training.

## F  TRAINING SCU VIA STOCHASTIC VARIATIONAL INFERENCE

From $\boldsymbol{\Theta} = (\mathbf{V}, \mathbf{W})$: $\mathbf{V}$ consists of $(\boldsymbol{\pi}, \mathbf{g})$ in CDs, we further rewrite $\mathbf{W}$ by $(\mathbf{W}^{\texttt{NC}}, \mathbf{W}^{\texttt{C}})$: $\mathbf{W}^{\texttt{NC}}$ the parameters in NCs, and $\mathbf{W}^{\texttt{C}}$ is the remaining ones. One can safely ignore the effect of $\mathbf{V}$ during training of $(\mathbf{W}^{\texttt{NC}}, \mathbf{W}^{\texttt{C}})$, since they remain fixed. Recall that each noise $\theta$ from a NC is assumed to follow $\mathrm{LogN}(\theta|\mu, \sigma^2)$. They can be re-written with a noise $\varepsilon$ from the standard normal distribution, i.e., $\theta = f((\mu, \sigma), \varepsilon) = \exp(\mu + \sigma \cdot \varepsilon)$, where $\varepsilon \sim \mathcal{N}(0, 1^2)$. In such case that each noise $\theta$ from NC can be "re-parametrized" with an non-parametric noise and the corresponding parameters $\phi = (\mu, \sigma)$, we can then use *stochastic variational inference* (Kingma & Welling, 2013) for the optimization of $(\mathbf{W}^{\texttt{NC}}, \mathbf{W}^{\texttt{C}})$ with a minibatch-based stochastic gradient method (see Appendix A for more details). Then, the final loss we minimize for a minibatch $\{(x_{i_k}, y_{i_k})\}_{k=1}^{M}$ becomes:

$$\mathcal{L}_{\texttt{SCU}}(\mathbf{W}^{\texttt{NC}}, \mathbf{W}^{\texttt{C}}) = -\frac{1}{M} \sum_{k=1}^{M} \log p(y_{i_k} | x_{i_k}, f(\mathbf{W}^{\texttt{NC}}, \varepsilon_{i_k}), \mathbf{W}^{\texttt{C}}) + \frac{1}{N} \sum_{(\theta, \phi)} D_{\mathrm{KL}}(q_\phi(\theta) \| p(\theta)) \quad (24)$$

where $\varepsilon_{i_k} = (\varepsilon_{i_k, u})_{u=1}^{|\phi|}$ is a sampled vector from the fully-factorized standard normal distribution, and $D_{\mathrm{KL}}(\cdot \| \cdot)$ denotes the KL-divergence. Although not shown in (24), an extra regularization term $R(\mathbf{W}^{\texttt{C}})$ can be added to the loss for the non-Bayesian parameters $\mathbf{W}^{\texttt{C}}$, e.g., weight decays.

In fact, in our case, i.e. $q_\phi(\theta) = \mathrm{LogN}(\theta|\mu, \sigma^2)$ and $p(\theta) = \mathrm{LogU}(\theta)$, $D_{\mathrm{KL}}(q_\phi(\theta) \| p(\theta))$ becomes improper:

$$D_{\mathrm{KL}}(\mathrm{LogN}(\theta|\mu, \sigma^2) \| \mathrm{LogU}(\theta)) = C - \log \sigma, \qquad \text{where} \quad C \to \infty. \quad (25)$$

As we explain in Appendix B, SBP bypasses this issue by using truncated distributions on a compact interval $[a, b]$ for $q_\phi(\theta)$ and $p(\theta)$. We found that, however, this treatment also imposes extra computational overheads on several parts of training process, such as on sampling noises and computing

$D_{\mathrm{KL}}(q_{\boldsymbol\phi}(\boldsymbol\theta)\|p(\boldsymbol\theta))$. These overheads are non-negligible on large models like ResNet or DenseNet, which we are mainly focusing on. Therefore, unlike SBP, here we do not take truncations on $q_\phi(\theta)$ and $p(\theta)$ due to practical consideration, assuming an approximated form between the truncated distributions of $q_\phi(\theta)$ and $p(\theta)$ on a large interval. Then we can replace each $D_{\mathrm{KL}}(q_\phi(\theta)\|p(\theta))$ in (24) by $-\log\sigma$ for optimization. In other words, each noise $\theta$ in NC is regularized to a larger variance, i.e., the more "noisy". We observed that this approximation does not harm much on the performance of SCU. Nevertheless, one should be careful that $q_\phi(\theta)$ and $p(\theta)$ should not be assumed as the un-truncated forms itself, but instead as approximated forms of truncated distributions on a large interval, not to make the problem ill-posed. As used in SBP, if they are truncated, the KL-divergence becomes:

$$D_{\mathrm{KL}}(\mathrm{LogN}_{[a,b]}(\theta|\mu,\sigma^2)\|\mathrm{LogU}_{[a,b]}(\theta)) = \log\frac{b-a}{\sqrt{2\pi e\sigma^2}} - \log\Phi(\beta) - \Phi(\alpha) - \frac{\alpha\phi(\alpha) - \beta\phi(\beta)}{2(\Phi(\beta) - \Phi(\alpha))}, \tag{26}$$

where $\alpha = \frac{a-\mu}{\sigma}$, $\beta = \frac{b-\mu}{\sigma}$, $\phi(\cdot)$ and $\Phi(\cdot)$ are the p.d.f. and c.d.f. of the standard normal distribution, respectively.

## G   EXPERIMENT SETUPS

**Datasets.** We perform our experiments extensively on CIFAR-10 and CIFAR-100 (Krizhevsky, 2009) classification datasets. CIFAR-10/100 contains 60,000 RGB images of size $32 \times 32$ pixels, 50,000 for training and 10,000 for test. Each image in the two datasets is corresponded to one of 10 and 100 classes, respectively, and the number of data is set evenly for each class. We use a common scheme for data-augmentation (Srivastava et al., 2015; Lin et al., 2013; He et al., 2016a; Huang et al., 2017b). ImageNet classification dataset, on the other hand, consists of 1.2 million training images and 50,000 validation images, which are labeled with 1,000 classes. We follow (Huang et al., 2017a; He et al., 2016a) for preprocessing of data in training and inference time.

**Training details.** All models in our experiments is trained by stochastic gradient descent (SGD) method, with Nesterov momentum of weight 0.9 without dampening. We use a cosine shape learning rate schedule (Loshchilov & Hutter, 2016), i.e., decreasing the learning rate gradually from 0.1 to 0 throughout the training. We set the weight decay $10^{-4}$ by for non-Bayesian parameters of each model. We train each CIFAR model for 300 epochs with mini-batch size 64 following Huang et al. (2017b), except for the "DenseNet-BC-190+*mixup*" models as they are trained for 200 epochs following the original setting (Zhang et al., 2018). For ImageNet models, on the other hand, we train for 120 epochs with mini-batch size 256.

**Configurations of SCU.** When a SCU $\mathbf{S} = (W^{\mathtt{NC}}, W^{\mathtt{CD}}, W^{\mathtt{BN}}, W^{\mathtt{Conv}})$ is employed in a model, we initialize $W^{\mathtt{NC}} = (\boldsymbol\mu, \boldsymbol\sigma)$ by $(\mathbf{0}, e^{-3})$, and $W^{\mathtt{CD}} = (\pi_i, g_i, \mathbf{b}_i)_{i=1}^I$ by $(i, 1, \mathbf{0})_{i=1}^I$. Initializations of $W^{\mathtt{BN}}$ and $W^{\mathtt{Conv}}$ may differ depending on models, and we follow the initialization scheme of the given model. In our experiments, we follow a pre-defined calling policy when `dealloc` and `realloc` will be called throughout training. If `dealloc` is used, it is called at the end of each epoch of training. On the other hand, if `realloc` is used, it start to be called after 10% of the training is done, called for every 3 epochs, and stopped in 50% of training is done. The thresholds for `dealloc` and `realloc`, i.e. $T_l$ and $T_h$, is set by 0.0025 and 0.05, respectively, except for CondenseNet-SCU-182 (Table 3), in which $T_l$ is adjusted by 0.001 for an effective comparison with the baseline. For all the CIFAR-10/100 models, we re-initialize $\mathbf{b}_i$ by a random sample from $[-1.5, 1.5] \times [-1.5, 1.5]$ pixels uniformly whenever a channel slice $\mathbf{S}_i$ is re-open via `realloc` process. We set the weight decay on each $\mathbf{b}_i$ to $10^{-5}$ separately from the other parameters. For the ImageNet results (Table 2), however, we did not jointly train $\mathbf{b}$ for faster training. Instead, each $\mathbf{b}_i$ is set fixed unless it is re-initialized via `realloc`. In this case, we sampled a point from $[-2.5, 2.5] \times [-2.5, 2.5]$ pixels uniformly for the re-initialization. We found that this simple re-allocation scheme can also improve the efficiency of SCU.

**Models.** In general, every model we used here forms a stack of multiple bottlenecks, where the definition of each bottleneck differs depending on its architecture (see Table 5). Each stack is separated into three (CIFAR-10/100) or four (ImageNet) stages by average pooling layers of kernel $2 \times 2$ to perform down-sampling. Each of the stages consists $N$ bottleneck blocks, and we report which $N$ is used for all the tested models in Table 6. The whole stack of each model follows a global average pooling layer (Lin et al., 2013) and a fully connected layer, and followed by single convolutional

layer (See Table 7). There exist some minor differences between the resulting models and the original papers (He et al., 2016a; Huang et al., 2017b; Xie et al., 2017). In ResNet and ResNeXt models, we place an explicit $2 \times 2$ average pooling layer for down-sampling, instead of using convolutional layer of stride 2. Also, we use a simple zero-padding scheme for doubling the number of channels between stages. In case of DenseNet, on the other hand, our DenseNet models are different from DenseNet-BC proposed by Huang et al. (2017b), in a sense that we do not place a $1 \times 1$ convolutional layer between stages (which is referred as the "compression" layer in the original DenseNet). Nevertheless, we observed that the models we used are trained as well as the originals.

Table 5: Listing of definition for each architecture block used in our experiments. Here, $\mathrm{BRC}_{I \to I'}^{K \times K}$ denotes $\mathrm{Conv}_{I \to I'}^{K \times K} \circ \mathrm{ReLU} \circ \mathrm{BN}$, $\mathrm{GConv}_{I \to I'}^{K \times K}$ denotes a group convolution with 4 groups with kernel size $K \times K$, and LGC denotes the learned group convolution layer originally proposed by Huang et al. (2017a).

| Architecture | $\mathcal{R}$ | $\mathcal{F}'$ | $\mathcal{W}$ |
|---|---|---|---|
| ResNet | $\mathrm{BRC}_{I \to I/4}^{1 \times 1}$ | $\mathrm{BRC}_{I/4 \to I}^{1 \times 1} \circ \mathrm{BRC}_{I/4 \to I/4}^{3 \times 3}$ | $\mathbf{X} + \mathbf{X}'$ |
| DenseNet | $\mathrm{BRC}_{I \to 4k}^{1 \times 1}$ | $\mathrm{BRC}_{4k \to k}^{3 \times 3}$ | $[\mathbf{X}, \mathbf{X}']$ |
| ResNeXt | $\mathrm{BRC}_{I \to I/2}^{1 \times 1}$ | $\mathrm{BRC}_{I/2 \to I}^{1 \times 1} \circ \mathrm{GConv}_{I/2 \to I/2}^{3 \times 3} \circ \mathrm{ReLU} \circ \mathrm{BN}$ | $\mathbf{X} + \mathbf{X}'$ |
| CondenseNet | $\mathrm{LGC}_{I \to 4k}^{1 \times 1}$ | $\mathrm{GConv}_{4k \to k}^{3 \times 3} \circ \mathrm{ReLU} \circ \mathrm{BN}$ | $[\mathbf{X}, \mathbf{X}']$ |

Table 6: Listing of $N$ values used for each model, with respect to the dataset that each model is trained. For ImageNet models, we use different values of $N$ for each stage.

| Dataset | Model | $N$ |
|---|---|---|
| CIFAR-10/100 | DenseNet-40 | 6 |
| | DenseNet-100 | 16 |
| | ResNet-164 | 18 |
| | ResNeXt-29 | 3 |
| | CondenseNet-SCU-182 | 30 |
| ImageNet | ResNet-50 | 3, 4, 6, 3 |
| | DenseNet-121 | 6, 12, 24, 16 |

Table 7: The generic model configurations used in our experiments with CIFAR-10/100 and ImageNet datasets. Here, $\mathrm{AvgPool}$ and $\mathrm{MaxPool}$ denotes the average pooling and the max pooling layer with kernel size $2 \times 2$ of stride 2, respectively. $\mathrm{GAvgPool}$ indicates the global average pooling layer, and $\mathrm{FullyConnected}$ indicates a fully-connected layer. Unless otherwise specified, the stride of the other operations are set to 1.

| CIFAR-10/100 | | ImageNet | |
|---|---|---|---|
| Module | Channel size | Module | Channel size |
| $\mathrm{Conv}^{3 \times 3}$ | $32 \times 32$ | $\mathrm{Conv}_{\mathrm{stride:2}}^{7 \times 7}$ | $112 \times 112$ |
| - | - | $\mathrm{MaxPool}$ | $56 \times 56$ |
| $\mathrm{Block} \times N$ | $32 \times 32$ | $\mathrm{Block} \times N_1$ | $56 \times 56$ |
| $\mathrm{AvgPool}$ | $16 \times 16$ | $\mathrm{AvgPool}$ | $28 \times 28$ |
| $\mathrm{Block} \times N$ | $16 \times 16$ | $\mathrm{Block} \times N_2$ | $28 \times 28$ |
| $\mathrm{AvgPool}$ | $8 \times 8$ | $\mathrm{AvgPool}$ | $14 \times 14$ |
| $\mathrm{Block} \times N$ | $8 \times 8$ | $\mathrm{Block} \times N_3$ | $14 \times 14$ |
| - | - | $\mathrm{AvgPool}$ | $7 \times 7$ |
| - | - | $\mathrm{Block} \times N_4$ | $7 \times 7$ |
| $\mathrm{AvgPool}$ | $1 \times 1$ | $\mathrm{AvgPool}$ | $1 \times 1$ |
| $\mathrm{FullyConnected}$ | - | $\mathrm{FullyConnected}$ | - |

