# OpenReview forum: "Selective Convolutional Units: Improving CNNs via Channel Selectivity"
_ICLR.cc/2019/Conference_

### Official Review · AnonReviewer2 · 2018-11-02
**promising idea but over-complicated method.**

**Rating:** 5
**Confidence:** 3

**Review:**

The main contribution of the paper are a set of new layers for improving the 1x1 convolutions used in the bottleneck of a ResNet block. The main idea is to remove channels of low “importance” and replace them by other ones which are in a similar fashion found to be important.  To this end the authors propose the so-called expected channel damage score (ECDS) which is used for channel selection. The authors have shown in their paper that the new layers improve performance mainly on CIFAR, while there’s also an experiment on ImageNet
It looks to me that the proposed method is overly complicated. It is also described in a complicated manner.  I don't see clear motivation for re-using the same features. Also I did not understand the usefulness of applying the spatial shifting of the so-called Channel Distributor. It is also not clear whether the proposed technique is applicable to only bottleneck layers.
The results show some improvement but not great and over results that as far as I know are not state-of-the-art (to my knowledge the presented results on CIFAR are not state-of-the-art). The results on ImageNet also show decent but not great improvement. Moreover, the gain in reducing the model parameters is not that great as the R parameters are only a small fraction of the total model parameters. Overall, the paper presents some interesting ideas but the proposed approach seems over-complicated

---

> ### Author Response · Authors · 2018-11-25
> **Response to AnonReviewer2**
>
> Many thanks for your time and effort to review our paper. We respond to your questions and concerns one-by-one in what follows. In addition, please check out the common response we have posted together, that addresses several important concerns raised by multiple reviewers in common.
>
> Q1. "I don't see clear motivation for re-using the same features."
> - The motivation for the re-using is to give important features more parameters. As explained in Section 2.1, notice that a 1x1 convolution performs nothing but a "pixel-wise linear transformation" on feature dimension, so that its parameters can be represented by a N x N’ matrix, provided that N=(# input channels) and N’=(# output channels). This implies that a single input channel is processed by 1 x N’ parameters when an input comes into the layer, and therefore re-using a feature n times implies that the feature is processed by n x N’ parameters.
>
> Q2. "I did not understand the usefulness of applying the spatial shifting."
> - As explained in Section 2.2 in more details, spatial shifting is a trick to properly utilize the re-used parameters. Considering again that n copies of a feature occupies n x N’ parameters of the matrix (Q1-1 above), one may notice that the naive copy would not help on expressivity of the convolution, since it is basically a linear transformation. Even though SCU contains ReLU inside the structure, this kind of phenomenon does happen during training. By using spatial shifting, we now can utilize the n x N’ parameters for "enlarging" the convolution kernel specially for the feature. Ablation study on spatial shifting demonstrated in Figure 3a clearly shows its effectiveness.
>
> Q3. "It is also not clear whether the proposed technique is applicable to only bottleneck layers."
> - We expect that the proposed method is still valid for other than bottleneck (it is mentioned in Section 4). Nevertheless, we primarily focus on the bottleneck setting under the presence of identity connection, because we expect this scenario is one of the best applications of channel-selectivity. In Section 2.1 of the revised draft, we provide more detailed intuitions and motivations why we study such bottleneck layers.
>
> Q4. “The gain in reducing the model parameters is not that great as the R parameters are only a small fraction of the total model parameters.”
> - The fraction of bottlenecks for the total parameters is NOT always small, and several state-of-the-art models invest very large portion of parameters on bottlenecks as follows:
>
>   (a) CondenseNet-SCU-182 model presented in Table 3 is a nice example of achieving high efficiency by exploiting its high fraction of bottlenecks. Initially, CondenseNet-SCU-182 has 6.29M parameters with 741M FLOPs in total before training the model. As reported in Table 3, these values can be reduced to 2.59M and 286M, respectively, and this reduction is mainly due to compression on the bottlenecks. In fact, this model invests 5.89M parameters only for bottlenecks, which is *93.7%* of the total parameters.
>
>   (b) DenseNet-BC-190, newly added in the revised draft as the state-of-art model also invests a lot of parameters for bottleneck, namely 17.5M (as reported in Table 1) out of 25.6M. In general, DenseNet models heavily rely on bottleneck structure for efficiency, and the overhead from the bottleneck itself becomes increasingly large as the model grows.
>
> As the examples demonstrate, reducing overhead from bottlenecks has been one of the crucial barriers for designing a large-scale, yet efficient CNN model.

---

### Official Review · AnonReviewer3 · 2018-11-05
**some concerns need to be clarified**

**Rating:** 5
**Confidence:** 3

**Review:**

This is an architecture design paper. It proposes a general structure called Selective Convolutional Unit that the authors claim to be useful for various CNN models. The SCU structure contains two major parts: CD and NC. CD for compressing/pruning channels and NC for multiplicative noise. The paper gives a measure, called expected channel damage score, on the change of the output for SCU. It also shows the effectiveness of SCU on CIFAR-10, CIFAR-100 and imagenet.

Some questions and concerns:

1. The paper spends too much space introducing the bottleneck structures and a whole lot of the details on the optimization of NC and CD are put in the appendix. I would suggest to reduce the section of introductory part and put a shorter version of appendix A and B to the main text so that the readers know more about the architecture and how it is optimized. In particular, the description on NC is confusing since without looking at the appendix it is not clear how the prior p(\theta) is used.

2. The experiment shows improvement on densenet and resnetXT, but the result is not the state-of-the-art. Wide-Resnet seems to get better accuracy on both CIFAR-10 and CIFAR-100 compared to the best accuracy reported in this paper. Also the number reported by the original densenet paper on imagenet seems to be better (densenet-264 has an error rate of 22.15/20.80)

3. In your CD design, channel assignment \pi is a discrete variable. How is it optimized in the training process?

4. The proof of proposition 1 does not look correct to me. The optimization procedure makes use of the data X to determine your NC variable \theta so \theta depends on X. In this way you cannot factorize the expectation in the equation below (20) in your appendix.

---

> ### Author Response · Authors · 2018-11-25
> **Response to AnonReviewer3**
>
> Many thanks for your time and effort to review our paper. We respond to your questions and concerns one-by-one in what follows. In addition, please check out the common response we have posted together, that addresses several important concerns raised by multiple reviewers in common.
>
> Q1. For your editorial suggestions
> - Many thanks for your thoughtful editorial suggestions. We revised the draft following them, where major revisions are marked by “red”.
>
> Q2. “How is the discrete \pi optimized in the training process?”
> - As we describe in Section 2.4, \pi is not directly trained via SGD, but updated via dealloc and realloc operations during training.
>
> Q3. “The proof of proposition 1 does not look correct to me.”
> - We remark that Proposition 1 does not involve anything related to optimization with X. Proposition 1 can be applied regardless on whether the network is trained or not (even in the case that the network is randomly initialized). Given that the network is fixed, \theta is completely independent on the distribution of X by design of NC, so that we can factorize the expectation.

---

### Official Review · AnonReviewer4 · 2018-11-14
**interesting idea, but really hard to read**

**Rating:** 6
**Confidence:** 2

**Review:**

This paper propose Selective Convolutional Unit (SCU), which can replace the bottleneck in Resnet block. The difference between SCU and bottleneck is that SCU adds Channel Distributor (CD) and Noise Controller (NC) to reduce and replace the channels. This paper also propose Expected channel damage score (ECDS) to measure the importance of a channel to decide weather remove or replace it. Then the experiment shows result on cifar10/100 and imagenet data set with different network architectures.
The idea is interesting, however, the parameter flops reduced rate seems not very impressive. The SCU seems too complicated,so I want to know that if the SCU could accelerate the forward process on modern GPU or mobile devices?
The result of these networks seems not the state-of-the-art, if the result can be improved, the SCU could be more convincing.

---

> ### Author Response · Authors · 2018-11-25
> **Response to AnonReviewer4**
>
> Many thanks for your time and effort to review our paper. We respond to your questions and concerns one-by-one in what follows. In addition, please check out the common response we have posted together, that addresses several important concerns raised by multiple reviewers in common.
>
> Q1. “The parameter flops reduced rate seems not very impressive.”
> - As we mentioned in our common response, “improving pruning efficiency” is not our major goal. Even though we place NC for inducing more sparsity on training (as pruning efficiency depends on training scheme), this is far from our key contributions, but closer to adopting Bayesian learning for better efficiency. Rather, we aim to explore a “safer way” of pruning and rewiring, without any post-processing after training. Namely, our goal is to achieve accuracy improvement and model compression simultaneously on a single pass of training. In this regard, our work is more about designing a new training scheme, complementary to any pruning works, for improving the network efficiency that allows some balancing between accuracy and compression on demand.
>
> Q2. “Can the SCU accelerate the forward process?”
> - Although implementing SCU for maximal acceleration is outside our scope, we expect that our method will do help on acceleration of networks at the inference time. Recall that SCU has two additional layers (NC and CD) compared to the standard BN-ReLU-Conv. The complexity from the layers, however, can be eliminated for those who need efficient inference:
>
>   (a) As mentioned in Section 2.4, noises in NC can be replaced by its expected values for faster inference. In fact, even the entire NC layer can be omitted by multiplying the expected values to the parameters in the former BN layer.
>
>   (b) In the case that SCU is trained using only dealloc (for compression), CD contains no spatial shifting so that the role of CD is nothing but re-indexing channels.
>
> Overall, the only expense of using SCU is a channel-selection layer via tensor indexing operation, while the remaining layers can work in a much smaller dimension in return. Comparative evaluations of CPU inference time in the table below show that SCU further improves the efficiency of CondenseNet-182 model through the training process, while outperforming other competitive models.
>
> CPU* inference time (per image)
> +-----------------------------------+-------------+--------------+----------+
> |          Model                        |   Before   |    After     |  Error  |
> |                                             |  training  |  training  |  rates  |
> +-----------------------------------+-------------+--------------+----------+
> | ResNeXt-29                        |        -        |  471.2ms | 3.58%  |
> +-----------------------------------+-------------+--------------+----------+
> | DenseNet-BC-250 (k=24) |        -        |  399.5ms | 3.64%  |
> +-----------------------------------+-------------+--------------+----------+
> | CondenseNet-SCU-182    | 114.8ms |*52.5ms* | 3.63%  |
> |                                              |                 | (-54.3%)   |             |
> +-----------------------------------+-------------+--------------+----------+
> *Intel Xeon E5-2630v4 @ 2.20GHz

---

### Author Response · Authors · 2018-11-25
**Common response to all the reviewers**

We sincerely thank all the reviewers for their valuable comments and effort to improve our manuscript. Major revisions in the new draft are temporarily colored by “red” for the reviewers’ convenience. In below, we provide a brief summary of the major revisions:

* Following the suggestion of AnonReviewer3, we have reduced the introductory part of bottleneck structure in Section 2.1. Instead, the space is devoted for describing how SCU is optimized in Section 2.4.

* Following the suggestion of AnonReviewer2/3/4, We have added more experimental results on the state-of-the-art level model (namely, DenseNet-BC-190) in Table 2.

* Based on the comment of AnonReviewer2, we have provided more detailed intuitions and motivations why we study the bottleneck structures in Section 2.1.

* Section 2.2 is significantly revised to help readers for better understanding of CD and NC.


In what follows, we respond several important concerns raised by multiple reviewers in common.

Q1. The proposed method is overly complicated (AnonReviewer2/4).
- We fully understand your concern, and revised the draft (marked by “red” texts) significantly for better understanding. Irrespectively, the proposed method, SCU, is highly-modularized and one can use it easily for any bottleneck CNN architectures (without any deep understandings on it). In particular, our PyTorch implementation of SCU works just like a standard convolutional layer, and one can convert an existing model into the SCU-counterpart by simply modifying <10 lines of code. We plan to open our source code after the paper decision, and we believe that this will further help the readers to understand the details of our method.

A way to understand our method better is to “separate” the complexity coming from NC apart from our method. This is because NC is a component for efficiency rather than our core idea. In other words, the complexity coming from NC can be replaced with other methods as long as they provide the efficiency. We revised the draft to emphasize this point (see the NC part of Section 2.2).

Q2. The results are not the state-of-the-art (AnonReviewer2/3/4).
- We believe that the effectiveness of our method is not limited to the model size. In the revision, we additionally perform a set of experiments on DenseNet-BC-190 with mixup augmentation [1], which showed the state-of-the-art level performance on CIFAR-10/100. The new results show that SCU is still beneficial to use, in a sense that: (a) S+D shows significant reduction of model parameters, (b) there is a meaningful improvement in accuracy from S+D to S+D+R, which indicates that realloc meaningfully utilized the pruned parameters, and (c) S+D+R consistently improves both accuracy and compression compared to the bare baseline, e.g., for CIFAR-10, we achieved reductions in (compression, error) = (-53.1%, -1.10%).

We emphasize again that our goal is neither to improve accuracy nor pruning alone. It is for improving the network efficiency that allows some balancing between accuracy and compression on demand. We believe it is an important problem rarely studied in the literature and we provide an important and novel step toward it.

[1] Hongyi Zhang, Moustapha Cisse, Yann N. Dauphin, and David Lopez-Paz. mixup: Beyond empirical risk minimization. In ICLR, 2018.

---

> ### Author Response · Authors · 2018-12-01
> **After First Rebuttal/Revision**
>
> Dear Reviewers and AC,
>
> We hope that all of you checked our rebuttal/revision and we would be very happy to answer any remaining questions/concerns.
>
> Thanks for your contribution to ICLR 2019,
> Authors

---

### Meta-Review · Area_Chair1 · 2018-12-16
**A new design for channel selection, yet over-complicated.**

**Confidence:** 4
**Recommendation:** Reject

**Metareview:**

This paper proposed Selective Convolutional Unit (SCU) for improving the 1x1 convolutions used in the bottleneck of a ResNet block. The main idea is to remove channels of low “importance” and replace them by other ones which are in a similar fashion found to be important. To this end the authors propose the so-called expected channel damage score (ECDS) which is used for channel selection. The authors also show the effectiveness of SCU on CIFAR-10, CIFAR-100 and Imagenet.

The major concerns from various reviewers are that the design seems the over-complicated as well as the experiments are not state-of-the-art. In response, the authors add some explanations on the design idea and new experiments of DenseNet-BC-190 on CIFAR10/100. But the reviewers’ major concerns are still there and did not change their ratings (6,5,5). Based on current results, the paper is proposed for borderline lean reject.